# Frustrated charge density wave and quasi-long-range bond-orientational order in the magnetic kagome FeGe

D. Subires[1,2], A. Kar[1], A. Korshunov[1], C. A. Fuller [3], Yi Jiang[1], H. Hu [1], D. Călugăru [4], C. McMonagle[3], C. Yi[5], S. Roychowdhury [5,6], W. Schnelle [5], C. Shekhar [5], J. Strempfer [7], A. Jana[8,9], I. Vobornik [8], J. Dai [10], M. Tallarida [10], D. Chernyshov [3], A. Bosak[11], C. Felser [5]✉, B. Andrei Bernevig[1,4,12] & S. Blanco-Canosa [1,12]✉

The intrinsic frustrated nature of a kagome lattice is amenable to the realization of exotic phases of matter, such as quantum spin liquids or spin ices, and the multiple-$\mathbf{q}$ charge density waves (CDW) in the kagome metals. Despite intense efforts to understand the mechanism driving the electronic modulations, its origin is still unknown and obscured by competing interactions and intertwined orders. Here, we identify a dimerization-driven 2D hexagonal charge-diffuse precursor in the antiferromagnetic kagome metal FeGe and demonstrate that the fraction of dimerized/undimerized states is the relevant order parameter of the multiple-$\mathbf{q}$ CDW of a continuous phase transition. The pretransitional charge fluctuations with propagation vector $\mathbf{q} = \mathbf{q}_M$ at $T_{CDW} < T < T^*$(125 K) are anisotropic, hence holding a quasi-long-range bond-orientational order. The broken translational symmetry emerges from the anisotropic diffuse precursor, akin to the Ising scenario of antiferromagnetic triangular lattices. The temperature and momentum dependence of the critical scattering show parallels to the stacked hexatic B-phases reported in liquid crystals and transient states of CDWs and highlight the key role of the topological defect-mediated melting of the CDW in FeGe.

The ground state of strongly degenerated frustrated lattices is a fertile ground for emergent phenomena driven by competing interactions[1,2]. For instance, the magnetic ground state and the long-range order of a frustrated network of spins is often a consequence of a subtle balance among the second (or higher order) nearest-neighbor and spatially anisotropic interactions[3,4], spin-orbit coupling[5], defects and disorder[6,7]. In strongly correlated electron systems with a high degree of frustration, Coulomb repulsion introduces interactions between spin, charge, and orbital degrees of freedom, providing a motivation for the study of competing intertwined orders[8,9].

Of particular interest is the phase transition in 2D triangular antiferromagnetic lattices[10,11], where spins are aligned 120° from each

[1]Donostia International Physics Center (DIPC), Paseo Manuel de Lardizábal, San Sebastián, Spain. [2]Departamento de Física Aplicada I, Universidad del País Vasco UPV/EHU, San Sebastián, Spain. [3]Swiss-Norwegian Beamlines at European Synchrotron Radiation Facility, Grenoble, Cedex, France. [4]Department of Physics, Princeton University, Princeton, NJ, USA. [5]Max Planck Institute for Chemical Physics of Solids, Dresden, Germany. [6]Department of Chemistry, Indian Institute of Science Education and Research Bhopal, Bhopal, India. [7]Advanced Photon Source, Argonne National Laboratory, Lemont, IL, USA. [8]CNR-Istituto Officina dei Materiali (CNR-IOM), Trieste, Italy. [9]International Center for Theoretical Physics (ICTP), Trieste, Italy. [10]ALBA Synchrotron Light Source, Barcelona, Spain. [11]European Synchrotron Radiation Facility (ESRF), Grenoble Cedex, Cedex, France. [12]IKERBASQUE, Basque Foundation for Science, Bilbao, Spain. ✉e-mail: Claudia.Felser@cpfs.mpg.de; sblanco@dipc.org

other in the basal plane, that unveils unconventional correlated diffuse patterns characteristic of frustrated magnetism[12,13], Kosterlitz-Thouless phases[14] or spin ices[15]. In the charge sector, the Kosterlitz-Thouless-Halperin-Nelson-Young (KTHNY) theory predicts that a 2D phase transition is topological[16–18], described by the continuous unbinding of topological defects, and the transition from an ordered solid to an isotropic liquid is commonly preceded by an intermediate state characterized by short-range positional but quasi-long-range bond orientational (BO) order[19–21].

The kagome lattice, a geometrically frustrated fabric of corner-sharing triangles[22], has recently emerged as a platform to study the phase transition from an electronic crystal (charge density wave, CDW) to an isotropic liquid. Due to the particular geometry of the kagome net that features van Hove (VHS) singularities, Dirac cones, and dispersionless flat bands[23], theory proposed the appearance of many body phases, allowing for the observation of anomalous and fractional Hall effect[24–27], chiral CDWs[28–30], superconductivity[31,32], loop currents[33–37] and heavy fermion physics[38–41]. At particular filling fractions, the Fermi surface is perfectly nested by a wavevector $q_M=(\frac{1}{2} 0)$, resulting in a $2 \times 2$ CDW. Examples of multiple-$q$ CDW orders have been observed in the $AV_3Sb_5$[42,43] (A = K, Rb) and $ScV_6Sn_6$[44] series of the kagome family. In the weakly correlated $AV_3Sb_5$[45,46], the first order phase transition is achieved without phonon softening[47], pointing to a prominent role of order-disorder scenarios[48,49]. In contrast, the ground state of $ScV_6Sn_6$ displays a different lattice landscape, with the collapse of a high-temperature soft mode at $\mathbf{q}^*=(\frac{1}{3} \frac{1}{3} \frac{1}{2})$[50,51] that competes with the low temperature ordered phase at $\mathbf{q}_{CDW}=(\frac{1}{3} \frac{1}{3} \frac{1}{3})$[52–57].

A different scenario is devised in the antiferromagnetic FeGe, holding the same lattice symmetry as AVS and SVS. Whereas the latter are nonmagnetic[58], FeGe orders antiferromagnetically (AFM) below ~400 K, with the magnetic moments aligned along the c-axis within each kagome layer and antiferromagnetically between planes (A-type antiferromagnetic order)[59,60]. A multiple-$q$ CDW strongly intertwined with the magnetic order develops below ~ 100 K with propagation vectors connecting the VHS at M, L and the AFM A points of the BZ, emphasizing the complex entanglement between charge, spin and lattice degrees of freedom[61–63]. The phase transition does not involve a phonon collapse at either M or L at $T > T_{CDW}$, but a sizable spin-phonon coupling of the low-energy mode at A[64,65] and a moderate hardening of an optical mode below $T_{CDW}$ at M[59]. On the other hand, several angle resolved photoemission spectroscopy (ARPES) experiments[66,67], not yet reproduced by density functional theory (DFT), highlighted the important role of both the orbital dependent saddle points close to the Fermi level[66] and trigonal Ge (Ge$_1$ in Fig. 1A) dimerization[64,66,68,69]. This debate has been further fueled by DFT calculations that pointed out a divergence of the electronic susceptibility, which correlates with the nesting function, at the K point of the BZ[70]. X-ray diffraction reported a dimerization of Ge$_1$ across the CDW[71] that is supported by the partial softening of the flat $k_z = \pi$ plane in the DFT+U calculations[64,72]. Furthermore, neutron and Raman scattering[73] found an enhancement of the crystalline symmetry upon cooling through the CDW transition and described the phase transition as an interplay between the L and A order parameters. Recent scanning tunneling microscopy (STM) experiments locate the CDW in the strong coupling regime[72], where phase fluctuations destroy the long range charge order and order-disorder scenarios might play a prominent role. However, following symmetry arguments, the dimerized scenario is also consistent with the presence of a primary order parameter at M and L and the nature of the multiple-$q$ CDW and its dynamics is far from being settled.

Here, we use x-ray diffraction, diffuse scattering (DS), ARPES, DFT, and Monte Carlo simulations to solve the symmetry and the electronic band structure of the low-temperature CDW phase of FeGe and to reveal a high-temperature quasi-2D hexagonal diffuse precursor localized along the M − L direction. The dimerization-induced lattice frustration demonstrates that the fraction of dimers can be considered as the relevant order parameter of the continuous phase transition. At intermediate temperature, $T_{CDW} < T < T^*(125\,K)$, we identify a state where the critical scattering is anisotropic around the M point, showing short-range positional but quasi-long-range bond orientational order, akin to the stacked hexatic phases observed in lyotropic liquid crystals[74]. Our results suggest that the phase transition fits within an order-disorder scenario captured by the Ising model of triangular

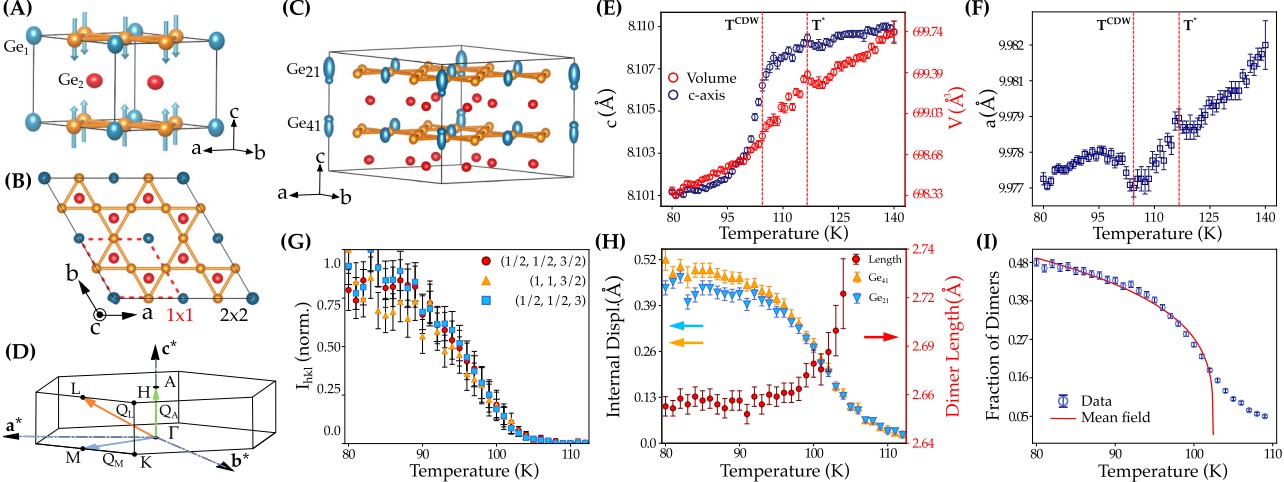

**Fig. 1 | X-ray diffraction and analysis of the order parameter. A** Normal-state structure (non-CDW) of FeGe and the spin polarization of the kagome planes. Orange symbols are the Fe atoms, the blue symbols represent the Ge atoms in the kagome plane (trigonal Ge$_1$) and the red symbols are the Ge atoms in the honeycomb layer (Ge$_2$). Arrows stand for the spin-up and spin-down in the Fe sites. **B** Top view of the dimerized CDW structure. **C** Dimerized CDW structure obtained from the hard x-ray refinement, highlighting the dimerization of the trigonal Ge$_1$ (here Ge$_{21}$ and Ge$_{41}$ move in opposite ways). The oval shape of Ge$_{21}$ and Ge$_{41}$ stand for their average site occupancy. **D** High symmetry points in the non-magnetic Brillouin zone. **E** Temperature dependence of the c-axis lattice parameter and volume,

V. **F** Temperature dependence of the a lattice parameter, highlighting the anomalies at T$^*$ and T$_{CDW}$. **G** Normalized temperature dependence of a set of CDW reflections. **H** Temperature dependence of the internal displacements of the trigonal Ge (Ge$_{21}$ and Ge$_{41}$) in the dimerized phase, showing a continuous 2nd order-like transition. The red axis refers to the temperature dependence of the dimer length (Ge$_{21}$-Ge$_{41}$ distance) in the dimerized phase. **I** Definition of fraction of dimers (fd) as the relevant order parameter and its fitting to a mean field power law, yielding a T$_{CDW}$ ~ 102 K. The departure from the mean field at T > T$_{CDW}$ is a consequence of the charge density fluctuations. The error bars represent the fit uncertainty.

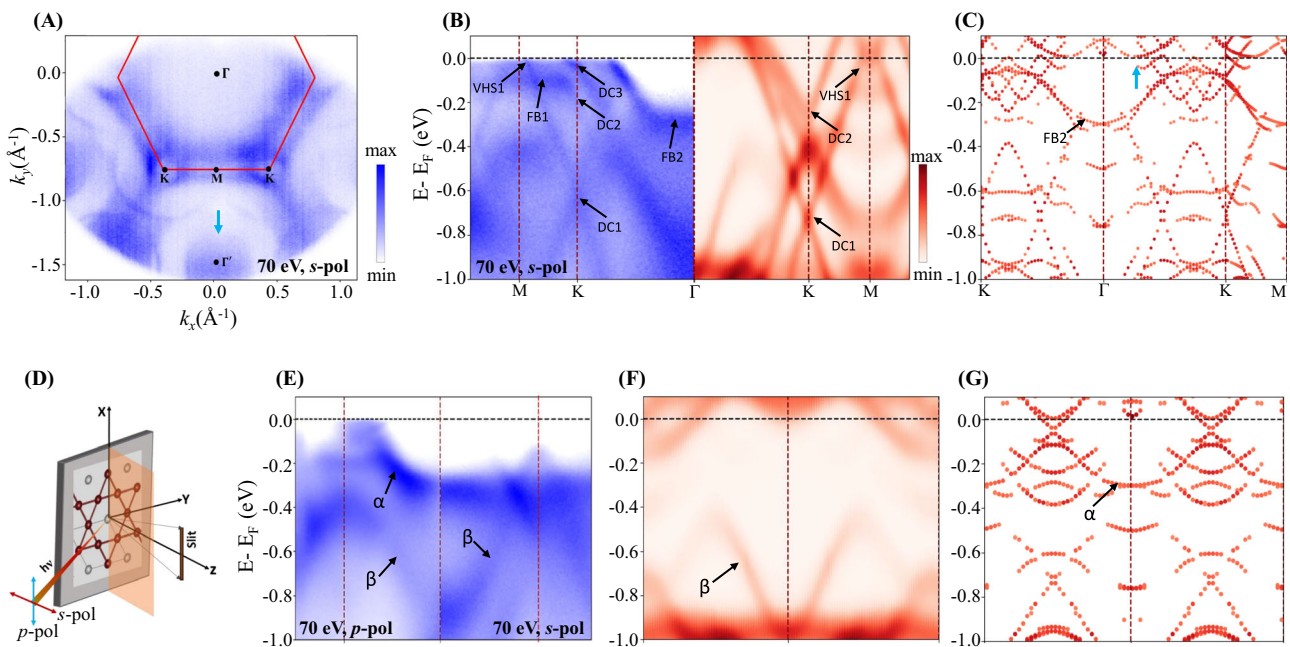

**Fig. 2 | ARPES and DFT calculations. A** Fermi surface map taken with $E_i = 70$ eV ($k_z = 0$) and s-polarized (s-pol) light at T = 10 K. **B** A comparative ARPES (blue) and DFT calculated bulk band structure (red) in CDW phase along Γ−K −M symmetry direction (calculated bulk band structures are Gaussian broadened with a broadening parameter of 0.06 eV). The ARPES spectra were taken with 70 eV photon energy with s-pol light. ('DC', 'VHS', and 'FB' represent the Dirac cone, Van Hove singularity, and flat band, respectively). **C** DFT calculated the folded surface band structure of FeGe along K− Γ− K–M with honeycomb surface termination in the CDW phase. **D** ARPES experimental geometry with the s-pol and p-pol light vectors. **E** M− Γ−M valence band spectra taken with 70 eV and p-pol and s-pol light (left and right panel, respectively). α and β stand for surface and bulk bands, respectively (see text). **F** Bulk calculated band structure along M− Γ−M with CDW phase. **G** M− Γ −M honeycomb surface terminated folded band structure of FeGe in CDW phase.

lattices[75] and the melting of the CDW is driven by the unbinding of topological defects, such as dislocation pairs and shear of domain walls[17,76,77].

# Results

## X-ray diffraction

Hexagonal FeGe (P6/mmm, space group No. 191) consists of individual FeGe kagome layers within the unit cell, with trigonal $Ge_1$, separated by honeycomb $Ge_2$ atoms (Fig. 1A, B)[78]. Figure 1E, F show the temperature dependence of the unit cell parameters obtained from the refinement of the x-ray diffraction patterns (see Supplementary Note 2). The unit cell volume shrinks ~0.2% between 140 and 80 K and identifies two critical temperatures; $T^* \sim 125$ K, more clearly visible in the thermal evolution of in-plane lattice parameter, and the CDW transition at $T_{CDW} \sim 105$ K. The change in volume shows a gradual crossover between two phases and is mostly driven by the shortening of the c-axis lattice parameter (~0.12%) (Fig. 1E) that smoothly varies through the transition, while the in-plane lattice parameters only undergo small structural variations at $T^*$ and $T_{CDW}$, Fig. 1F. The shortening of the c-axis is consistent with the dimerization of trigonal Ge in the kagome plane, as previously reported on a basis of magnetic energy saving and x-ray diffraction[64,68,71].

With further cooling below $T_{CDW}$, a multiple-**q** CDW develops with propagation vectors $\mathbf{q_M} = (\frac{1}{2} 0\, 0)$, $\mathbf{q_L} = (\frac{1}{2} 0\, \frac{1}{2})$ and $\mathbf{q_A} = (0\, 0\, \frac{1}{2})$. The temperature dependence of several normalized CDW peaks is summarized in Fig. 1G. The rather linear T-dependence of their intensities down to 80 K evidences that the phase transition cannot be properly identified as first-order. This growth in intensity following a continuous phase transition is also consistent with the predominantly second order character of the transition observed in the specific heat[59,71] (see Supplementary Fig. 1). We have indexed the low-temperature CDW phase structure within the non-centrosymmetric P6mm space group with a partial dimerization of the trigonal $Ge_1$, resulting in a disordered composite final structure - an overlay of dimerized and undimerized

regions[71]. We also point out that the low-temperature structure can also be indexed within the high-temperature centrosymmetric space group P6/mmm with a similar figure of merit ($R_1$) (see Supplementary Note 2), but non-centrosymmetric space groups were inferred to explain the double cone magnetic transition at low temperature[79]. At T = 80 K, we find that the dimerized trigonal Ge atoms (labelled as $Ge_{21}$ and $Ge_{41}$ in Fig. 1C) are lifted ~ ± 0.5 Å from the kagome plane, Fig. 1H, in agreement with the DFT calculations[68] and previous diffraction studies[64,71], and a dimerization fraction (the occupancy ratio of $Ge_1$ in a dimerized and undimerized position) of 50%, larger than the previously reported. Figure 1H displays the dimer length ($Ge_{21}$ and $Ge_{41}$ distances between adjacent kagome planes) that smoothly decreases below $T_{CDW}$ and reaches a constant value of 2.66 Å below 95 K. Following this experimental evidence, we define the fraction of dimers (fd), given by the occupancy of corresponding $Ge_1$ sites, as a relevant order parameter of the continuous phase transition, Fig. 1I. The gradual growth of fd can be fitted to a mean-field behavior, returning a critical temperature $T_{CDW} \sim 102$ K and a long tail of critical fluctuations, characteristic of a reduced dimensionality. The presence of short-range charge correlations at $T > T_{CDW}$ is reminiscent of the magnetic critical scattering in Ho thin films undergoing a dimensionality crossover[80].

## ARPES and DFT calculations

Having structurally characterized the FeGe crystals, we now move on to its electronic structure. In Fig. 2A, we show the Fermi map of FeGe obtained for $k_z = 0$ ($E_i = 70$ eV, T = 10 K) that partially covers both the first and second Brillouin zones. The band structure is in agreement with the previous experimental reports[59,66,67,69], showing a hexagonal Fermi surface typical of the kagome metals[78,81]. Our first observation is the sizable photoemission matrix element effects in different momentum spaces, highlighted by the surface state pocket emerging at Γ' of the neighboring BZs (Fig. 2A). The constant energy contours

have hexagonal symmetry with rounded triangular electron pockets surrounded by a larger circular hole pocket at each K point of the BZ. The triangular pockets result in Dirac crossings (DC3) around the Fermi level, while the circular hole pocket corresponds to van Hove singularity (VHS1), as labeled in Fig. 2B (see also Supplementary Fig. 3). The DFT orbital projections of the bands (see Supplementary Figs. 8 and 9) show that the Fermi surface presents mainly Fe-3d orbital character and reproduces the quasi-flat bands, VHS, and Dirac points close to the Fermi level[62,65,66,70]. The dimerized trigonal $p_z$ orbital of $Ge_1$, which mainly contributes to the electron pocket at the Brillouin zone center Γ in the normal state, is pushed down −0.5 to −1.0 eV below the Fermi level in the CDW state (see Supplementary Figs. 10 and 11) and is not visible in Fermi surface contour.

Focusing on the $k_z = 0$ plane, in Fig. 2B, E, we plot the energy-momentum band dispersion of FeGe along Γ-K-M and Γ-M high symmetry directions, respectively. In Fig. 2B (left panel), three different Dirac crossings (DC1, DC2, and DC3) are identified and located at −0.05, −0.15, and −0.65 eV below the Fermi level at the corner of the BZ. The measured Dirac velocity for DC1 is $v_{DC1} = 2.7 \times 10^5$ m/s, which is comparable to $AV_3Sb_5$[81]. A flat band at −0.12 eV below the Fermi level appears along the K−M direction (FB1), consistent with the AFM spin majority band reported by DFT calculations[66]. Another flattish band behavior was observed around the Γ point along the Γ-K symmetry direction, −0.28 eV below $E_F$ (FB2), assumed to be responsible for the correlated electronic properties in FeGe[62]. In Fig. 2E, a comparative ARPES spectrum taken with p-polarized (p-pol, left panel) and s-polarized (s-pol, right panel) light along Γ-M high symmetry direction is shown for $k_z = 0$ plane. The U-shaped band around Γ is visible for p-pol light, indicating an in-plane orbital contribution to the band structure. Additionally, the '∧' shaped band observed at M the point, 0.1 eV below $E_F$ (Fig. 2E, right panel) is visible for s-pol light and is attributed to the $d_{xz}$ and $d_{yz}$ orbitals (see Supplementary Fig. 10). Along the Γ-M direction, two VHS were identified at M close to the Fermi level, more clearly visible along the K-M-K direction using circularly polarized light (see Supplementary Fig. 3). Overall, the kagome features are consistent with the previous ARPES experiments but its comparison with the ab-initio DFT simulations had relied on individual renormalization factors of the electronic band dispersion[59,66], which were not explained or derived by DFT calculations.

Aided with the P6mm symmetry of the CDW state, Fig. 2 (Supplementary Figs. 3 and 4) compares the energy dispersion of FeGe bands for $k_z = 0$ plane with the DFT calculations considering the bulk (Fig. 2 (B right panel) and F) and folded CDW surface bands (Fig. 2C, G) with honeycomb termination along the Γ-K-M and Γ-M high symmetry directions, respectively. From Fig. 2, we observe that the ARPES band structure of FeGe is an admixture of the bulk and surface states. Without any adjustable parameter nor the inclusion of the Hubbard term (U), the band dispersion along the Γ-K-M direction agrees with the DFT simulations, both in electron velocity and bandwidth, resulting in Dirac crossings (DC1) mainly derived from the $d_{xy} - d_{x^2-y^2}$ orbitals of Fe (Fig. 2 B, Supplementary Figs. 10 and 13). The V-shaped band observed at −1 eV binding energy at Γ, matched by a CDW bulk band, originates mainly from the $p_z$ orbitals of trigonal $Ge_1$ that hybridize with the $d_{xz}-d_{yz}$ orbitals of Fe and is dragged down from -0.5 to -1 eV in the CDW phase (β band in Fig. 2E and F). Furthermore, the U-shaped band at −0.28 eV at Γ is attributed to CDW honeycomb surface folded bands ($d_{xz} - d_{yz}$ orbitals of Fe) with contribution from the CDW bulk bands ($d_{xz}-d_{yz}$ orbital character of Fe) on $k_z = \pi$ plane after folding, i.e., folded from L ($\frac{1}{2}$ 0 $\frac{1}{2}$) (α band in Fig. 2E) (see also Supplementary Figs. 5 and 10). Therefore, surface U-shaped bands can also be seen as $k_z$-projected bulk bands with some surface reconstructions. Moreover, the '∧' shaped band 0.1 eV below $E_F$ observed at M, Fig. 2E, is a result of a combination of bulk (close to Γ) and surface bands (close to M). The larger agreement between the experimental and ab-initio band structure and the complete orbital description of the kagome bands,

without the need for any renormalization factor, downplay the correlation effects to describe the electronic band structure of FeGe.

## Diffuse scattering and Monte Carlo simulations

With the crystal and electronic structure of CDW-FeGe solved in detail, we now pay attention to the role of the M, L, and A high symmetry points of the BZ in the formation of the charge density wave. In particular, we aim to search for diffuse signals at $T > T_{CDW}$ that are characteristic fingerprints of local (short-range) pretransitional fluctuations of a CDW phase transition, crucial to identify the leading instabilities[82].

First, let us focus on the (h 0 l) plane of the kagome lattice, whose temperature dependence diffuse map is displayed in Fig. 3A. Clear diffuse clouds, already visible at room temperature (RT), are present at $h \pm \frac{1}{2}$ and $l \pm \frac{1}{2}$ for h and l integer. The diffuse scattering (DS) develops sizable intensity around the l = 2 region of the reciprocal space, with a rod-like diffuse intensity along the M-L direction (yellow ellipses in Fig. 3A), resulting in a stack of nearly uncorrelated kagome layers and a very short-range 2 × 2 order. The diffuse intensities are strongly modulated along the l-direction, due to the negative correlation of the atomic displacements along the c-axis, presumably associated with the $Ge_1$ dimerization. The diffuse precursor is nearly temperature independent upon cooling down to $T^* = 125$ K. Below $T^*$, the spectral weight starts to accumulate at the L point, namely at ($\frac{1}{2}$ 0 $\frac{3}{2}$) and ($\frac{3}{2}$ 0 $\frac{3}{2}$), and smoothly increases, diverging at 105 K, in agreement with the transition temperature observed by x-ray diffraction in Fig. 1F. The diffuse precursor is completely absent in the in-plane polarized AFM FeSn[83] (see Supplementary Fig. 27), demonstrating that the origin of the correlated rod-like diffuse signals is a result of the dimerization-driven short-range charge fluctuations arising from the out-of-plane AFM-coupled spins in a triangular lattice.

The charge precursor also localizes at the M point below ~125 K, following the same temperature dependence as at the L point. The DS down to 105 K, fitted to a Lorentzian function, returns a value of the out-of-plane correlation length of less than one unit cell for both M and L. Remarkably, no DS is detected at (0 0 $\frac{1}{2}$) (A point in the BZ) from RT down to $T_{CDW}$. Following the $Ge_1$ dimerization and the spin-phonon coupling mechanisms reported experimentally and theoretically[64,68] as the main driving force for the CDW formation, the absence of any pre-transitional scattering at the A point of the BZ seems to contradict those scenarios. On general grounds, one would expect an enhancement of the charge density correlation function driven by a progressive magnetostriction-driven dimerization of the kagome planes that eventually collapses in a superlattice reflection at $T_{CDW}$, as observed experimentally in spin-Peierls compounds[84,85]. With further cooling, CDWs with propagation vectors (0 0 $\frac{1}{2}$) (A), ($\frac{1}{2}$ 0 0) (M) and ($\frac{1}{2}$ 0 $\frac{1}{2}$) (L) appear at $T_{CDW} = 105$ K with an in- (out-of) plane correlation lengths of A = 40 ± 3 Å (40 ± 3 Å), M = 43 ± 2 Å (41 ± 2 Å) and L = 40 ± 3 Å (38 ± 2 Å), respectively, in good agreement with the values reported in the literature[72].

Next, we investigate the temperature dependence diffuse maps in the (h k 2) plane in Fig. 3B that shows even richer diffuse features (see Supplementary Fig. 20 for the (h k 3) plane). At RT and down to ~125 K, distinctive DS holding a 6-fold symmetry emerges in the form of a hexagonal diffuse pattern with a diameter of $\mathbf{a}^* \sim 0.72$ Å$^{-1}$, surrounding the Bragg reflections. The diffuse intensity does not vary either with the azimuthal angle or from ring to ring but with l. Besides, the independence of the diffuseness with (h k) for any l indicates a highly ordered crystalline structure. This structured DS in momentum space is a hallmark of the strong geometric frustration of a triangular lattice[12,13], arising from the out-of-plane AFM-coupled kagome planes, hence imaging a fabric of emergent dimerized/non-dimerized clusters in the normal state.

Between 125 K and $T_{CDW}$, an anisotropic diffuse signal starts to condense at the M point of the BZ ($\frac{1}{2}$ 0 0) together with a strong dependence of the hexagonal diffuse intensity in the (h k 2) plane, characteristic of occupational (or substitutional) disorder (dimerized-

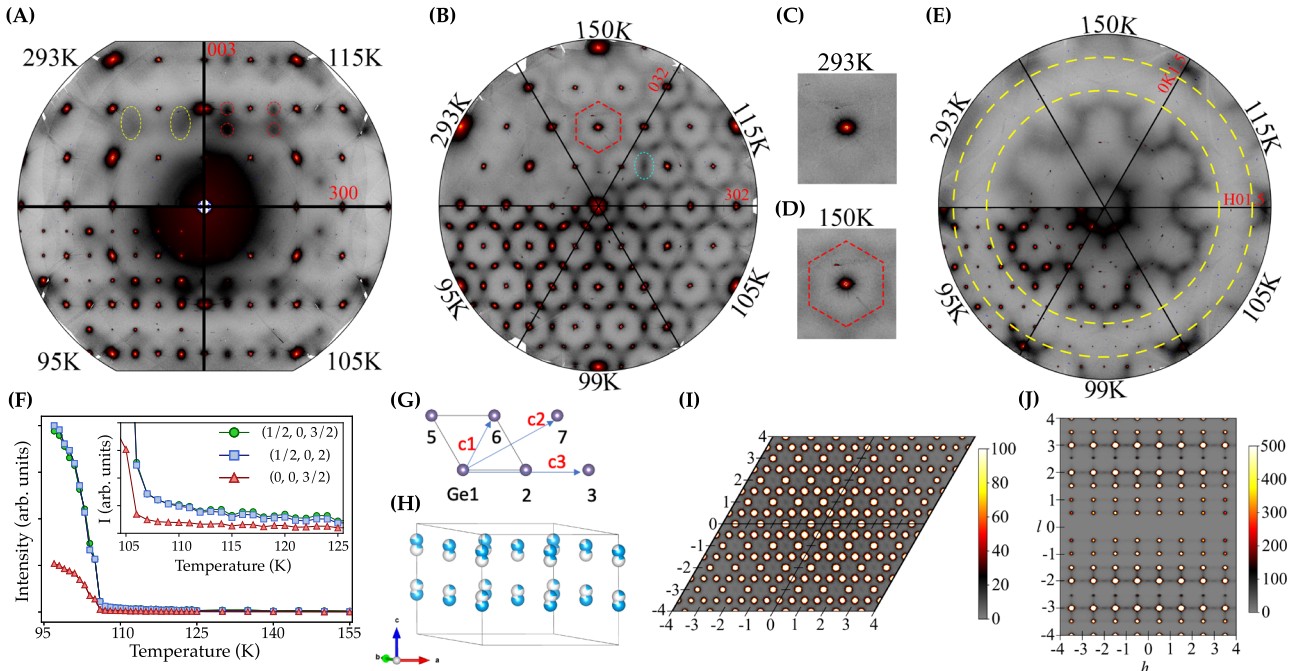

**Fig. 3 | Diffuse scattering and Monte Carlo modeling. A** (h 0 l) Diffuse maps at different temperatures. The yellow ellipse at 300 K highlights the DS along the M-L direction and demonstrates the uncorrelated charge scattering between kagome layers. No charge precursor is detected at the A point. **B** (h k 2) Diffuse maps as a function of temperature. The red hexagon underlines the 2D hexagonal shape of the DS. At 115 K, the anisotropic charge scattering concentrates at the M points (blue ellipse). **C, D** Zoom-in of the hexagonal DS at 293 K and 150 K. **E** (h k $\frac{3}{2}$) diffuse maps. The absence of scattering enclosed in between the yellow circumferences is a result of the small in-plane Fe displacements. **F** Temperature dependence of the DS

as a function of temperature obtained from the integration of a region of interest (ROI) defined as red circles in (**A**). Inset, zoom-in of the DS between the temperature interval 105 K < T < 125 K. **G** Schematics of the model used for the MC simulations. $c_i{}'s$ describe the interaction energies taken into account in the Ising Hamiltonian. $c_4$ stands for the out-of-plane nearest-neighbor interaction. **H** Real-space configuration of the average structure of the $Ge_1$ included in the MC simulation. **I, J** (h k 2) and (h 0 l) DS maps obtained from MC, respectively, indexed according to the room temperature disordered unit cell.

non dimerized states), rather than driven by atomic vibrations, whose scattering goes to zero at low q (see Supplementary Note 7). With further cooling, the anisotropic diffuse signal smoothly grows in intensity down to 105 K, thus preserving the $C_6$-symmetry of the lattice, in agreement with the x-ray diffraction data. Focusing on half integer values of l, the (h k $\frac{3}{2}$) plane in Fig. 3E, also reveals a honeycomb diffuse pattern, whose intensity strongly varies with the azimuthal angle and q (see Supplementary Fig. 20 for the (h k $\frac{3}{2}$) and (h k $\frac{5}{2}$) cuts), presumably driven by the small in-plane displacements of the Fe atoms[86]. Some quasi-circular hexagonal shape of the DS is still present below $T_{CDW}$, hence differentiating from Brazovskii scenario reported in isotropic systems[87], skyrmion lattices[88], and hole doped cuprates[89].

Visualizing the hexagonal DS as a composite of binary disorders that try to pack in a triangular lattice of P6mm symmetry[90], the substitutional disorder can be modeled by a triangular Ising antiferromagnet lattice. To fully understand the 3D diffuse pattern, we have carried out Monte Carlo (MC) simulations to achieve a microscopic realization of a large 2 × 2 × 2 unit cell (see Supplementary Note 8), starting from a negative nearest neighbour correlation and assuming that the $Ge_1$ at (0.5 0 z), (0.5 0.5 z) and (0 0.5 z) are disordered. In the MC simulation, dimerized and non-dimerized $Ge_1$ were modeled according to the Ising Hamiltonian:

$$H = \sum_{<i,j>:NN} c_1 \sigma_i \sigma_j + \sum_{<i,j>:NNN} c_2 \sigma_i \sigma_j$$
$$+ \sum_{<i,j>:4NN} c_3 \sigma_i \sigma_j + \sum_{<i,j>:z-NN} c_4 \sigma_i \sigma_j + \sum_i h\sigma_i + E_0, \quad (1)$$

- where $c_{i=1,2,3}$ are in-plane nearest-neighbour (NN), next nearest-neighbour (NNN) and 4th-NN coupling, $c_4$ is z-directional NN coupling, h is the magnetic field and $E_0$ is a constant. $\sigma$ is a 'spin' value (−/+)1

representing dimerized and non-dimerized $Ge_1$, respectively. The DS simulations for the (h k 2) and (h 0 l) planes are displayed in Fig. 3I−J, nicely matching the experimental diffuse maps. Furthermore, the 3D model also reproduces the Bragg nodes at the A point of the BZ as a result of the doubling of the unit cell (see Supplementary Fig. 30). The high degree of frustration that emerges upon cooling and the absence of DS at the A point indicates that the dynamics of the CDW in FeGe is of order-disorder type and further confirms that the fraction of dimerized states can be considered as the relevant order parameter. We note that our model (Eq. (1)) is *derived* from ab initio studies of the dimerization energetics, and *not* just fitted to the data. This is a distinct new method, presented in Supplementary Note 8, which allows a further check on the consistency of our results.

## Anisotropic DS and quasi-long-range BO order

Focusing on the intermediate phase between $T^*(125 K) > T > T_{CDW}$, the DS at the M point develops a sizable anisotropic peak broadening, see Fig. 4A, B, not observed at the L point. The diffuse profile at 125 K was fitted to a Lorentzian lineshape[91–93] and its correlation length extends to 5 Å along $q_∥$ (Γ-M-Γ direction, radial peak width) and less than one unit cell along $q_⊥$ (K-M-K direction, azimuthal peak width) directions, respectively, indicating a short-range positional but quasi-long-range bond orientational order of the sixfold director. This anisotropic scattering is characteristic of the melting of the CDW by topological defects that appear due to thermal fluctuations in the 2D kagome plane, akin to the stacked hexatics observed in multilayer smectic B phases of liquid crystals[94,95]. The temperature dependence of the peak widths (Fig. 4D) follows a mean field behavior, while the peak width ratio $(q_∥/q_⊥)$ (Fig. 4E), which is proportional to the mean dislocation distance, approaches to ~1 at $T_{CDW}$, as expected in proximity to a broken translational symmetry state. It is

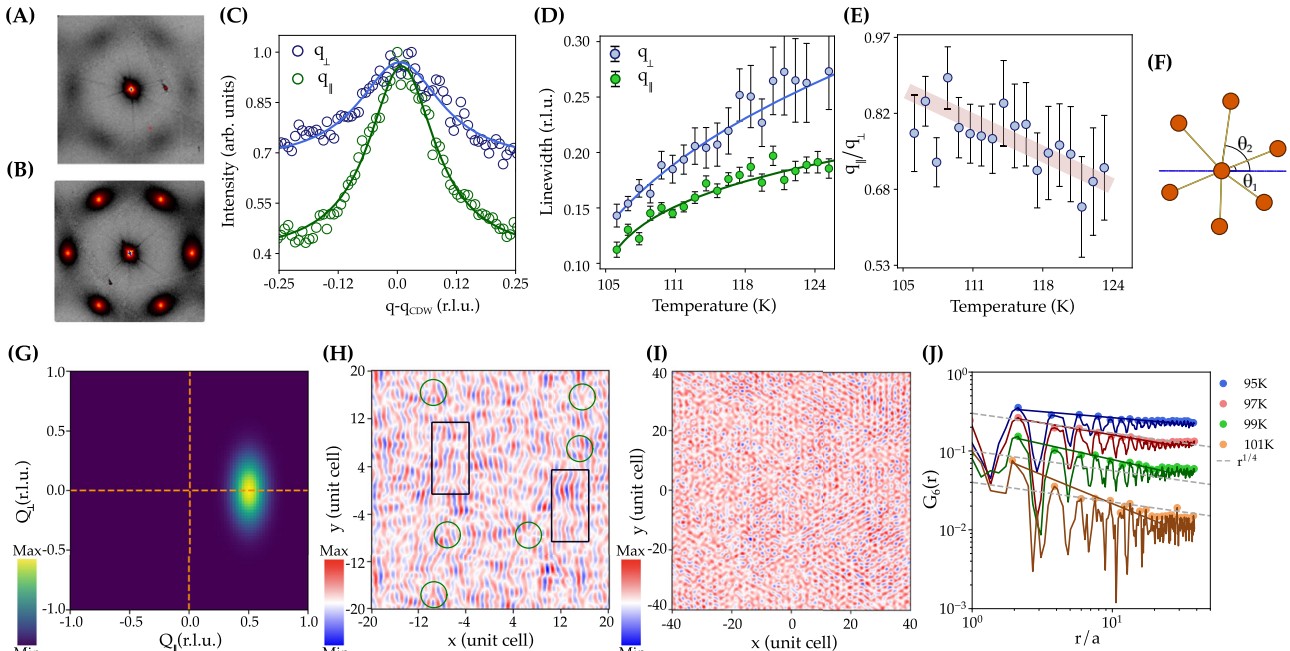

**Fig. 4 | Anisotropic DS and quasi-long-range BO order. A** DS map around the (1 1 2) Bragg peak at T = 115 K. **B** T = 95 K. **C** Profile of the diffuse signal along the Γ-M-Γ (parallel to $q_M$, $q_\parallel$, radial direction) and the K-M-K (perpendicular to $q_M$, $q_\perp$, azimuthal directions). **D** Temperature dependence of the DS peak width, $q_\perp$ and $q_\parallel$. The solid line is fitting to a critical power law. **E** Temperature dependence of the peak width ratio, $q_\parallel/q_\perp$. The faded line is a guide to the eye. **F** Sketch illustrating the bond orientational order parameter $\Psi_6(\mathbf{r}_i)$ defined by the Eq. (2). **G** Simulated DS of a single-q CDW in a triangular lattice with the experimental peak width at 95 K. **H** Fourier Transform (FT) of the simulated single-q DS of (**G**). The melting of the CDW is driven by dislocation unbinding (green circles) and shear (black boxes). **I** FT of the experimental triple-q CDW of FeGe at 95 K, showing a real space reconstruction of the charge density. The charge density map was discretized by the Voronoi tessellation to find the neighbor statistics of the kagome lattice (see Supplementary Note 9). **J** The bond-order correlation function, $G_6(r)$ shows a constant value ~ 1 at T = 95 K, as expected in the solid phase, and an algebraic decay at 95 K < T < $T_{CDW}$, that locates the solid to hexatic transition. Solid lines are fits of the upper envelopes of the data to an algebraic decay ~ $r^\beta$. A universal critical value of $\eta_6 \to \frac{1}{4}$ (grey line) is predicted by the KTHNY theory on approaching the hexatic to the liquid phase. The error bars represent the fit uncertainty.

also worth noting that some anisotropy is still present below $T_{CDW}$, suggesting a coexistence of regions with short and long-range translational order[86]. This hints at the fragility of the low-temperature CDW state in FeGe[72], further supported by the drastic reduction of the anisotropy of DS in annealed crystals (see Supplementary Figs. 20 and 22)[96,97]. In Fig. 4G, we simulate the DS in the reciprocal space for the anisotropic broadening of the single-q diffuse signal observed experimentally. The anisotropic spatial profile of the $q_M$ charge scattering is consistent with the melting of the CDW by defects (dislocations in the green-dashed circles) and shear (black boxes) in Fig. 4H with a reduced coherence of the 1D chain. However, when the CDW peak broadens parallel to $q_M$, the real space maps describe a relative phase change along the 1D domain walls (see Supplementary Fig. 31), akin to the formation topological defects in transition metal dichalcogenides and superconductors[56,98–100].

## Discussion

The results presented here bring important information for the understanding of the multiple-q CDW transition in FeGe and in kagome metals in general. First of all, within both the equally plausible low-temperature centrosymmetric P6/mmm and non-centrosymmetric P6mm space groups, we can reach a better description of the band structure at the DFT level without either the assumption of renormalization factors or the Hubbard term, U, in DFT. This has important consequences for the lattice dynamics calculations and puts constraints on the U values used to soften a rather flat phonon mode in the $k_z = \pi$ plane[64]. It appears that the electronic structure of FeGe is less correlated than previously assumed, despite the strong intertwining of charge and spin orders. Indeed, the comprehensive description of the band structure must rely on the precise treatment of dimerized/undimerized crystal structure and the AFM within the DFT framework[69,70,79,86,101].

Moreover, we have identified a quasi-2D hexagonal diffuse scattering along the M-L line of the BZ at high temperatures that evolves towards a localized charge precursor at M and L points at low temperatures. Although, in principle, this would discard charge fluctuations at A as the leading instability and the driving force of the CDW, the strong geometric frustration of the triangular lattice introduced by the dimer formation actually causes the DS at M and L points, demonstrating an order-disorder transformation that follows a continuous phase transition. Nevertheless, we point out that the order-disorder scenario in FeGe differs from the $AV_3Sb_5$, where the phase transition is achieved by the freezing, without softening, of a transverse phonon mode[47,49]. More importantly, the short-range charge fluctuations are also reminiscent of a fragile metastable CDW phase or some glassiness that would explain the extreme sensitivity to external perturbations[72] and the effect of annealing[67].

On the other hand, although the profile and correlation length of the diffuse spots demonstrate the presence of phase fluctuations, bearing a strong resemblance to critical scattering due to the melting of dislocation pairs and shear, we caution about setting the phase transition within the KTHNY theory. The continuous topological melting via unbinding of defect-pairs is predicted to occur in 2-dimensional systems[18], thus, the precise character of the phase transition requires a topological analysis of the real space charge density. This is further justified since FeGe is a 3D system, although the $FeGe_1$ kagome plane is purely 2D, and the melting should occur as a single, first-order transition.

Within the KTHNY theory[17], the bond orientational (BO) order parameter is parametrized by a local ordering field describing

orientation between neighboring sites:

$$\Psi_6(\mathbf{r}_k) = \frac{1}{N_k} \sum_{j=1}^{N_k} e^{i6\theta_{kj}}, \qquad (2)$$

- where $N_k$ represents the maxima of the charge density distribution (equivalent to the number of particles in a colloidal system) around a reference point located at position $\mathbf{r}_k$ and $\theta_{kj}$ defines the angle the $k$-$j$ bond, Fig. 4F. The bond-order correlation function $G_6(r)$ is defined as:

$$G_6(r) = \frac{1}{N_r} \sum_{<i,j>}^{N_r} \Psi_6(\mathbf{r}_i)\Psi_6^*(\mathbf{r}_j), \qquad (3)$$

- where $N_r$ is the charge density at a distance $r$. The KTHNY theory states that $G_6(r)$ decays algebraically in the hexatic phase, $G_6(r) \propto r^{-\eta_6}$, indicating quasi-long-range bond order, and exponentially in the liquid phase, $G_6(r) \propto e^{-r/\xi_6}$, where $\xi_6$ is the orientational correlation length.

Figure 4I shows the Fourier transform (FT) of the corresponding structure factor $S(\mathbf{q})$ at T = 95 K[83,102–104]. The FT, representing the real space distribution of the charge density, is further discretized by performing a Voronoi tessellation (see Supplementary Note 9). As displayed in Fig. 4J, $G_6(r)$ is close to 1 at 95 K, and its upper envelope can be fitted to a constant, as expected in the solid phase. Increasing the temperature results in a faster algebraic decay, demonstrating the quasi-long-range orientational order up to 101 K that would define the solid-to-hexatic transition, but, nevertheless, does not follow with an exponential decay, as the KTHNY theory predicts on approaching the liquid phase. This is a consequence of the Voronoi construction that does not capture in detail the charge density discretization in real space at T > 105 K and prevents the extraction of the Frank's constant, $K_A$, that describes the effective stiffness of the BO field, and a comprehensive analysis of the BO order parameter.

In conclusion, we have carried out a comprehensive experimental survey to show that the CDW in the antiferromagnetic FeGe is of order-disorder type that fits within the Ising model of a triangular lattice. The order-disorder transformation is a direct consequence of the strong frustration introduced by the dimerization of the trigonal $Ge_1$ on the kagome plane that double the unit cell along **a**, **b** and **c** directions of the crystal. Moreover, we observe an anisotropic diffuse signal for temperatures T > $T_{CDW}$, characteristic of the melting of the CDW by topological defects that resemble the hexatic B phases in liquid crystals and TMDs[56,98,99]. Microscopically, the anisotropic DS could be a consequence of the fragility of the CDW or driven by the small in-plane Fe displacements, as inferred by x-ray diffraction[86] and infrared spectroscopy[105]. Although our correlation analysis is consistent with a quasi-long-range bond orientational order at T ≤ $T_{CDW}$, it cannot guarantee that FeGe fits within the KTHNY theory. However, it opens new perspectives to look at the melting of the charge modulations in kagome lattices and the possibility of studying them in detail in the 2D limit[106] with more advanced diffraction techniques.

## Methods

Single crystals of FeGe were grown by the chemical vapor transport method using iodine as a transport agent. High-purity Fe and Ge powders were mixed together with a molar ratio of Fe:Ge = 1:1. The mixture and iodine (~10 mg/cm³) were loaded into a quartz tube and vacuum sealed. The tube was placed in a two-zone furnace, ΔT= 620–560 °C[105].

Single crystal diffraction was carried out at the Swiss-Norwegian beamline (SNBL) BM01, European Synchrotron Radiation Facility (ESRF), with incident energy $E_i$ = 20 keV and a Pilatus 2 M detector[107].

The raw data were processed with SNBL toolbox and CrysAlis Pro software. The refinements were carried out with SHELXL2018/1 using SHELXLE as the GUI. Single crystal diffuse scattering was performed at the ID28 beamline at ESRF with $E_i$=17.8 keV and a Dectris PILATUS3 1M X area detector. We use the CrysAlis software package for the orientation matrix refinement and the ID28 software ProjectN for the reconstruction of the reciprocal space maps and plotted in Albula. The components (h k l) of the scattering vector are expressed in reciprocal lattice units (r.l.u.), (h k l)= h$\mathbf{a}$* + k$\mathbf{b}$* + l$\mathbf{c}$*, where $\mathbf{a}$*, $\mathbf{b}$*, and $\mathbf{c}$* are the reciprocal lattice vectors.

Hard x-ray resonant scattering experiments at the Fe-K edge ($E_i$= 7.115 keV) were performed at the beamline 4ID-D of the Advanced Photon Source at Argonne National Laboratory.

Angle Resolved Photoemission spectroscopy experiments (ARPES) were performed at the LOREA beamline (MBS electron analyzer, base pressure of $10^{-10}$, angular resolution of 0.2°, energy resolution 10 meV) of ALBA and APE-LE beamline (DA30 electron analyzer, base pressure of $10^{-10}$ mbar, angular resolution of 0.2°, energy resolution 10 meV) of ELETTRA research facilities.

The first-principle calculations in this work use the Vienna ab initio Simulation Package (VASP)[108–112] with generalized gradient approximation of Perdew-Burke-Ernzerhof (PBE) exchange-correlation potential[113]. A 8 × 8 × 8 (5 × 5 × 5) k-mesh for non-CDW (CDW) phase and an energy cutoff of 500 eV are used. The maximally localized Wannier functions are obtained using WANNIER90[114–117]. A local coordinate system at the kagome site is adopted in order to decompose $d$ orbitals when construct MLWFs, the same as the one used in Ref. 62. The Wannier tight-binding models are symmetrized using *Wannhr_symm* in *WannierTools*[118]. The unfolding of CDW bands is performed using *VaspBandUnfolding* package[119,120]. The Fermi surface is computed using *WannierTools*[118] and visualized using *Fermisurfer*[121].

Monte Carlo (MC) simulations were performed to generate real-space realizations of possible atomic configurations that could be used to model the observed diffuse scattering. The average FeGe unit cell at 80 K, solved from the single crystal diffraction experiments on BM01, was taken and expanded to a 32 × 32 × 32 supercell with all trigonal Ge atoms set to be non-dimerized (represented in the simulation as the Ising variable, $\sigma = +1$). A small number (~5%) of the Ge was randomly converted to dimers (represented with $\sigma = -1$). In each MC move, an Ising variable was randomly inverted, and the moves were accepted or rejected following the Metropolis condition with the Hamiltonian expressed in eq.(1) and the MC temperature set to 80 K. Four key interactions between neighboring Ising variables were considered: the nearest neighbor (NN), next nearest neighbor (NNN), and the third nearest neighbors (3NN) in the ab-plane ($c_1$, $c_2$, and $c_3$, respectively), and the NN along the c-axis, $c_4$. The parameter $h$ is a magnetic consideration that accounts for the overall proportion of dimerized Ge in the model. The numerical values for $c_1$, $c_2$, $c_3$, $c_4$, $h$ and $E_0$ were derived from DFT calculations. Five MC simulations were run in this way, all starting from different randomized configurations to try to minimize the possibility of getting stuck in a local energy minimum. Each simulation was run until the energy converged, and then the resulting atomic configuration was used as input to the program Scatty[122], which calculates the average diffuse scattering from all given simulations. The small in-plane displacement of the Fe atoms is not included in the MC simulations, hence, the DS does not reproduce the absence of diffuse intensity within the dashed-yellow area in Fig. 3E.

## Data availability

The scattering, ARPES, DS, and IXS data generated in this study can be accessed from https://doi.org/10.6084/m9.figshare.c.7731683.v1.

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

## Acknowledgements

We acknowledge Ming Yi, Yu He, E. da Silva-Neto, A. Subedi, D. Efremov, C. Schüssler-Langeheine, A. Frano, Stephen Wilson, Philippe Bourges, Hu Miao, Riccardo Comin, and Peter Keim for fruitful discussions and critical reading of the manuscript. D.S., A.Kar and S.B-C. acknowledge financial support from the MINECO of Spain through the project PID2021-122609NB-C21 and by MCIN and by the European Union Next Generation EU/PRTR-C17.I1, as well as by IKUR Strategy under the collaboration agreement between Ikerbasque Foundation and DIPC on behalf of the Department of Education of the Basque Government. A.K. thanks the Basque government for financial support through the project PIBA-2023-1-0051. Y.J and H.H. were supported by the European Research Council (ERC) under the European Union's Horizon 2020 research and innovation program (Grant Agreement No. 101020833) as well as by IKUR Strategy. D.Căl. acknowledges the hospitality of the Donostia International Physics Center, at which this work was carried out. D.Căl. was supported by the European Research Council (ERC) under the European Union's Horizon 2020 research and innovation program (grant agreement no. 101020833) and by the Simons Investigator Grant No. 404513. B.A.B. was supported by the Gordon and Betty Moore Foundation through Grant No.GBMF8685 towards the Princeton theory program, the Gordon and Betty Moore Foundation's EPiQS Initiative (Grant No. GBMF11070), Office of Naval Research (ONR Grant No. N00014-20-1-2303), Global Collaborative Network Grant at Princeton University, BSF Israel US foundation No. 2018226, NSF-MERSEC (Grant No. MERSEC DMR 2011750), the Simons theory collaboration on frontiers of superconductivity, and Simons collaboration on mathematical sciences. C.F. acknowledges support from the DFG under SFB 1143 (Project No. 247310070), the Würzburg-Dresden Cluster of Excellence on Complexity and Topology in Quantum Matter - ct.qmat (EXC 2147, Project No. 390858490), and FOR 5249 (QUAST, Project No. 449872909). This work has been partly performed in the framework of the nanoscience foundry and fine analysis (NFFA-MUR Italy Progetti Internazionali) facility. LOREA beamline is co-funded by the European Regional Development Fund (ERDF) within the ´Framework of the Smart Growth Operative Program 014-2020´. This research used resources from the Advanced Photon Source, a U.S. Department of Energy (DOE) Office of Science User Facility operated for the DOE Office of Science by Argonne National Laboratory under Contract No. DE-AC02-06CH11357.

## Author contributions

S.B-C. conceived and managed the project. C.Y., S.R., C.S., and C.F. synthesized the single crystals. W.S. measured the specific heat and magnetization. D.S., A.Kar., A.J., I.V, J.D., M.T., and S.B.-C. conducted the ARPES experiments. D.S. and A.Kar analyzed the ARPES data. A.K. and A.B. measured the D.S. and C.Fuller performed the Monte Carlo and the DS simulations. J.S. and S.B-C. carried out the resonant X-ray scattering experiments. C.M., D.Chernyshov, D.S., A. Kar, and S.B.-C. took the hard x-ray diffraction, and D.C. refined and solved the structure. H.H., Y.J., D.Călugăru, and B.A.B. carried out the DFT calculations. D.S. and S.B-C. simulated the diffraction patterns and the correlation analysis. S.B.-C. wrote the manuscript with input from all coauthors.

## Funding

## Competing interests

The authors declare no competing interests.
