## [Transparent Peer Review file · Nature Communications]

Frustrated charge density wave and quasi-long-range bond-orientational order in the magnetic kagome FeGe

Corresponding Author: Dr Santiago Blanco-Canosa

Version 0:

Reviewer comments:

Reviewer #1

(Remarks to the Author)

The authors present a combined ARPES and diffuse scattering study of the kagome magnet FeGe. Specifically, they find a charge-diffuse precursor in the antiferromagnetic phase that indicates the formation of Ge dimerization well above the charge density wave ordering temperature. They draw analog to the hexatic B-phases reported in liquid crystals and transient states of the CDWs, and points out the potential role of topological defects in the melting of the CDW order. This is a very rich and scholarly work that deserves to be published in Nature Communications. The main content as well as the supplementary materials are spectacularly rich, and will be of great interest to those working on kagome metal systems, of not only FeGe, but also of other well-known systems with charge density wave orders including 135 and 166, and even other CDW systems beyond kagome lattices. Therefore I recommend publication of this paper in Nature Communications. Below I list a number of comments for the authors to consider, which may help to improve the already great paper. None of the comments are critical but only for further improvements that I can suggest. Hence it is left to the discretion of the authors of which to address or not for revision.

Annealed crystals have more isotropic DS compared to as grown crystals. Since the authors state that the annealed crystals have longer range CDW, I assume this is the case where the CDW transition is higher? Can the authors provide this information of what the transition is for the annealed crystals? Also, does this mean that the isotropic DS indicates a better condition for the long-range CDW order? Can the authors comment on this? What happens to the disorder distribution in the annealed crystals if that is the cause of the anisotropic DS in the as-grown samples?

The authors describe the phase above T_{CDW} as a melting of the CDW that resembles a hexatic phase in liquid crystals. Do I understand correctly that this is analogous to the nematic phase above the SDW order in the iron-pnictides? And if so, can the analog be carried over further in that in the FeSCs, the nematic phase and SDW order occur simultaneously in the parent compound of certain material families and are only separated by doping? Here in FeGe, with annealing when the defects are presumably reduced and the DS is more isotropic, this phase above TCDW is also reduced? Is there evidence for this in the annealed crystals?

The authors make a point that FeGe is not correlated because DFT can capture all the features. Yet in the SM it is also stated that "However, since the system has a large number of bands near the Fermi level, it shows a heavily broadened spectrum where the discrepancy in the detailed features between theory and experiments is expected." Can the authors reconcile these statements? Also, another way to gauge the overall correlation strength is to compare the measured and calculated bandwidth of the Fe 3d valence bands. If the authors have data on this (like a large energy range comparison), it may be a good idea to include a comparison. If that data is not available, it is not critical.

Some even more minor points to consider:

It is stated that "surface state pocket emerging at Gamma' of the neighboring BZs". Can the authors label on the map which is the surface state?

While the DC1, 2 and 3 are all labeled in the measured data, could the corresponding features be also labeled in the calculated dispersion in Fig. 2B as well? Naively when I compared these two side by side, while the number of Dirac cones and even possibly the dispersion slopes are reproduced by calculations, the locations do not appear to match. It would help the readers if the corresponding features are labeled for a better comparison.

In similar veins, for the ease of the readers who would like to follow the discussion of the measured dispersions on pages 4 and 5, it may be helpful to refer to some of the bands with labels such as alpha, beta, etc, directly labeled on the figures.

Figure caption for Fig. 1: H and I should be combined according to the figure.

Figure caption for Fig. 3B: the blue ellipse mentioned seems to be missing in the figure.

Fig. S19: the caption says both annealed and as-grown samples are shown. But I'm not sure if the figure shows both, or only the annealed crystals since the temperature are all different?

For the FeGe_{0.9} sample shown in the SM, what is the transition around 200K from the magnetization?

Reviewer #2

(Remarks to the Author)

In the manuscript by Subires and colleagues, they studied the charge density wave (CDW) ordered phase in the antiferromagnetic kagome metal FeGe using x-ray diffraction (XRD), angle-resolved photoemissions spectroscopy (ARPES), density functional theory (DFT), and diffuse scattering. They first characterized lattice parameters as a function of temperature by XRD, identifying two characteristic temperature scales, namely T^* and TCDW. After the identification, they defined a relevant order parameter, the fraction of dimers given by the occupancy of corresponding Ge1 sites. Next, they carried out ARPES experiments and compared experimentally obtained electronic structures with their DFT calculation results. These results showed a good match, and the authors claimed that correlation effects may not be significant, contrary to previous studies. Finally, they presented diffuse scattering results at various temperatures together with anisotropy in diffuse scattering, showcasing evidence of short-range positional but quasi-long-range band orientation order. Ultimately, they described the nature of the CDW order in FeGe as an order-disorder scenario which follows a second-order like phase transition.

This is a comprehensive study, combining diffraction, diffuse scattering, and spectroscopy with theoretical calculations. The Supplementary Information is also extensive, providing valuable experimental and theoretical information on FeGe. However, they still need to confirm their main arguments with recent experimental reports, particularly on annealed FeGe. While they presented some of data sets from 300 °C-annealed FeGe in the Supplementary Information, more data are needed. I would like to ask authors to include datasets from various annealed FeGe, for examples, annealing temperature variation as shown in Wu et al., Phys. Rev. Lett. 132, 256501 (2024). Before I decide whether to recommend this manuscript for publication in Nature Communications, I would like to confirm if authors' scenario remains valid after these additional verifications. I outlined some critical points (leaving minor points aside for now) below..

1. They mentioned that normalized CDW peak intensity from XRD showed a linear T-dependence down to 80 K, implying that the CDW transition cannot be first-order, further supported by the small release of specific heat. However, a recent study on annealed FeGe, which has the long-range CDW order, shows significant peak structure at TCDW in the specific heat characterization (arXiv:2307.07990), strongly supporting a first-order like nature of the CDW transition. How can this discrepancy be resolved?
2. In their ARPES results, they did not find any signatures of a gap opening, which contradicts with the preprint on annealed FeGe (arXiv:2404.0223). This points also needs to be resolved. Furthermore, termination dependence (i.e., kagome & honeycomb) should be examined to fully address surface-originated bands in ARPES.
3. The authors downplayed the role of correlation effects based on good matches between the ARPES and DFT results with accurate consideration on the space group. I find this statement unconvincing, as they also mentioned, "Besides the aforementioned features that have a good matching between DFT bands and ARPES, we also observe some features that cannot be well matched." Those two statements are confusing due to their self-inconsistency. In addition, considering renormalization factors for DFT calculations are quite common for transition-metal-based systems (e.g., Fe-based superconductors), supported by comparisons between experimental angle-integrated valence band spectra and density of states calculations. It is hard to believe that FeGe is uniquely exempt from this general behavior of transition-metal-based systems.

Reviewer #3

(Remarks to the Author)

I have revised the present paper submitted to Nature Communications. The work contains a thorough analysis of the CDW transition in FeGe regarded as a Kagome metal. The authors elucidate the origin of such transition which they ascribe to interlayer Ge-Ge dimerization occurring below TCDW. They characterize the transition by introducing an Ising model describing the Kagome lattice degrees of freedom (dimerized vs. non-dimerized sites).

One strength of the paper which I highly value is the combination of several experimental (X-ray diffraction, diffuse scattering, ARPES) and theoretical techniques (DFT, Monte Carlo techniques) on the same material. In this way the authors are able to reach a solid conclusion on the origin of the CDW transition observed.

Since there is no consensus yet on the mechanism driving the CDW phase in FeGe, I believe that the present paper is timely, interesting and the observations reported relevant to understanding the physics of Kagome metals in general.

Nevertheless, I believe that some issues can be tightened further in the paper before being published. For instance, it would be useful if the present scenario for the CDW transition is more readily set in context comparing with previous descriptions. In particular, I miss the link with topological phenomena reported previously. More detailed comments can be found below.

Main issues/comments:

-There are indications from previous experimental work (Teng et. al., Nature 609, 490 (2022)) of an anomalous Hall effect occurring within the CDW state when $T < T_{CDW}$. I am wondering how relevant is this to the CDW formation from your experimental observations. Do you find any indications of such AHE? If that is the case how do you think model (1) should be extended to take account of such phenomena?

-One important message from the paper seems to be the small role played by electronic correlations in the CDW transition observed in FeGe. However, the large room temperature values of resistivity observed in FeGe as compared to those of Av_3Sb_5 (Teng et. al., Nature 609, 490 (2022)) seem to suggest that FeGe can be regarded a bad metal (or close to a bad metal). Such bad metal behavior is characteristic of strongly correlated materials. Isn't there then an inconsistency between your conclusions and the resistivity observations?.

-I miss an explanation of how model (1) is derived. For instance, which is the relation between the c_i and microscopic quantities of the material?

-The point is made that the CDW is caused by the dimerization between Ge atoms on different layers. It is not clear to me if this transition is actually driven by the electrons which order in the Kagome lattice planes which then can lead to the Ge dimerization observed. In such case the Coulomb repulsion between electrons on the Kagome lattices may drive the CDW instability. And then as a consequence of this the lattice distortion may occur. Could this scenario be also consistent with your observations?

-STM experiments (J.-X. Yin, et. al., PRL 129, 166401(2022)) display a suppression of the differential conductance at the Fermi energy when entering the CDW for $T < T_{CDW}$. However, your ARPES data indicate that no gap opening occurs at the CDW transition. Can you explain the two seemingly different observations?

-There is also evidence for the existence of an edge state in the CDW phase as observed in STM measurements (J.-X. Yin, et. al., PRL 129, 166401 (2022)). This edge state is of topological origin and is related to the topological character of the CDW state (and the AHE observed). The edge state is found to disappear at temperatures $T > T_{CDW}$. Is your analysis consistent with the existence of such edge state? Could it be related to the absence of a gap in the CDW as observed in ARPES?

Version 1:

Reviewer comments:

Reviewer #1

(Remarks to the Author)

I have read through the authors' response to all the referee comments and the revised manuscript. I am satisfied with the revision and recommend publication in Nature Communications.

Reviewer #2

(Remarks to the Author)

In the revised manuscript by Subires and colleagues, they have addressed almost all the concerns raised in the previous report. However, the response regarding annealed FeGe remains somewhat lacking, as they have not provided specific heat dataset in the response. Given that inconsistency in specific heat results could mean significant sample dependence, an explicit dataset for specific heat must be presented.

Additionally, during the review process, a preprint (arXiv:2410.13994) has been available online, providing experimental evidence on how annealing affects FeGe crystals. From their 4D-STEM results, Ge deficiency could be responsible for the absence of the CDW order in annealed samples at 550~560 degs. Among the authors' results from FeGe_{0.9}, the magnetic susceptibility closely resembles that of FeGe annealed at 550~560 deg. However, as they assigned a CDW transition for FeGe_{0.9}, and I wonder if they could provide a consistent explanation for the new results.

The last note is that more papers have been available online during the review (arXiv:2410.18063, arXiv:2411.10931, and arXiv:2412.18824), and some of them might be relevant and worth to be included in the manuscript.

Reviewer #3

(Remarks to the Author)

I have revised the response of the referees to my comments. The authors have covered in detail all the issues I have raised in their new version of the manuscript. In particular, the connection of their results to previous studies regarding the topological aspects of the transition is now clarified. The main points in their response can be summarized as follows:

-The authors have checked that they observe the AHE. They explain that such AHE is related with non-trivial topology of the Fermi surface but not with an order-disorder transition.

-The bad metallic state is due to disorder fluctuations rather than electron correlations.

-The origin of the CDW is most likely a magnetostriction effect.

-The opening of a gap in the CDW cannot be discarded due to the 20meV resolution in their ARPES measurements in contrast to STM measurements which have a much better resolution. Hence, the two measurements may consistently find a gap within the CDW.

Hence, my main concerns have been adequately taken into account. Also the point regarding the strength of electronic correlations has been clarified.

Based on the above I can recommend publication of the paper in Nature Communications.

We thank the referee's comments and their feedback that individually allowed us to improve the manuscript and the communication of our results. In the following, we are answering one by one all the questions/comments.

Reviewer #1 (Remarks to the Author):

The authors present a combined ARPES and diffuse scattering study of the kagome magnet FeGe. Specifically, they find a charge-diffuse precursor in the antiferromagnetic phase that indicates the formation of Ge dimerization well above the charge density wave ordering temperature. They draw analog to the hexatic B-phases reported in liquid crystals and transient states of the CDWs, and points out the potential role of topological defects in the melting of the CDW order. This is a very rich and scholarly work that deserves to be published in Nature Communications. The main content as well as the supplementary materials are spectacularly rich, and will be of great interest to those working on kagome metal systems, of not only FeGe, but also of other well-known systems with charge density wave orders including 135 and 166, and even other CDW systems beyond kagome lattices. Therefore, I recommend publication of this paper in Nature Communications.

- We appreciate the referee for his/her critical read of the manuscript and his/her enthusiasm towards our work. Indeed, we also believe that the results we present are of important interest to the community, not only working on kagome materials but also in CDW materials in general.

Below I list a number of comments for the authors to consider, which may help to improve the already great paper. None of the comments are critical but only for further improvements that I can suggest. Hence it is left to the discretion of the authors of which to address or not for revision.

- Annealed crystals have more isotropic DS compared to as grown crystals. Since the authors state that the annealed crystals have longer range CDW, I assume this is the case where the CDW transition is higher? Can the authors provide this information of what the transition is for the annealed crystals? Also, does this mean that the isotropic DS indicates a better condition for the long-range CDW order? Can the authors comment on this? What happens to the disorder distribution in the annealed crystals if that is the cause of the anisotropic DS in the as-grown samples?

- We have annealed two more samples at 440°C and 550°C and the data is plotted in the SI. The crystal with the highest T_{CDW} is the one annealed at 300°C, as reported also by other groups. It is interesting to note that the T_{CDW} of the 300°C annealed crystal is similar to T^* in the as-grown FeGe, hinting at the effect of reducing the intermediate phase with quasi-long-range orientational order. On the other hand, annealing at 400°C and 550°C gives a different result. The 400°C sample has a T_{CDW} of ~100K, similar to the as-grown crystal. The hexagonal rings are still present, although with less intensity, and the presence of orthorhombic domains is largely reduced. Comparing the 300°C and 400°C annealed FeGe, it seems that the isotropy of the CDW peak correlates with the larger correlation length of the peak, hence the referee is correct in his assumption. On the other hand, the FeGe sample annealed at 550°C presents a very intense disorder-driven hexagonal rings, without the presence of an anisotropic precursor. We believe that this sample is not representative for the study of the CDW in FeGe as it contains impurities of cubic FeGe that develops a transition at ~300 K, see figure below, hence increasing the level of extrinsic disorder.

The supplementary information has been updated with the new data and results.

Temperature dependence of the magnetic susceptibility of FeGe. The magnetic transition around 300 K is presumably associated to the formation of a skyrmion lattice.

- The authors describe the phase above T_{CDW} as a melting of the CDW that resembles a hexatic phase in liquid crystals. Do I understand correctly that this is analogous to the nematic phase above the SDW order in the iron-pnictides? And if so, can the analog be carried over further in that in the FeSCs, the nematic phase and SDW order occur simultaneously in the parent compound of certain material families and are only separated by doping? Here in FeGe, with annealing when the defects are presumably reduced and the DS is more isotropic, this phase above T_{CDW} is also reduced? Is there evidence for this in the annealed crystals?

- We thank the referee for this interesting point. The hexatic phase is a state of matter that is between the solid and the isotropic liquid phases in a two dimensional system and contains a six-fold orientational order, in analogy with the nematic phase (with two-fold orientational order). Indeed, in the 135 kagome family (CsV_3Sb_5 , KV_3Sb_5 , $CsTi_3Bi_5$), a nematic order has been reported by means of STM and ARPES, respectively. So, yes, they are analogous. Once the crystals are annealed (300-400°C), this hexatic phase is reduced and the anisotropic DS is hidden by the high temperature ring of DS (see the new data in the SI).

On the other hand, one would make the analogy between the nematic phase and SDW in FeSCs tuned by doping and the hexatic phase tuned by annealing in FeGe, but, to be honest, we would refrain about extending the analogy deeply. Fe pnictides are SC and the energy scales between the structural transition and the SDW are similar. In case of FeGe, these energy scales are rather separated ($T_N=440K$, $T_{CDW}=105K$). Besides, we believe that the quasi-long-range bond orientational order in FeGe is a consequence of some in-plane Fe vibration or *glassiness* and not driven by spin fluctuations.

- The authors make a point that FeGe is not correlated because DFT can capture all the features. Yet in the SM it is also stated that “However, since the system has a large number of bands near the Fermi level, it shows a heavily broadened spectrum where the discrepancy in the detailed features between theory and experiments is expected.” Can the authors reconcile these statements? Also, another way to gauge the overall correlation strength is to compare the measured and calculated bandwidth of the Fe 3d valence bands. If the authors have data on this (like a large energy range comparison), it may be a good idea to include a comparison. If that data is not available, it is not critical.

- We thank the referee for pointing out this apparent inconsistency. One of the main conclusions of our work is the better agreement of DFT and ARPES, without implying the use of large values of U or renormalization factors, but cannot fully capture all the details of ARPES, presumably due to the band broadening or the need for the proper treatment of dimerized/undimerized phases at low temperature. In other words, what our ARPES+DFT says is that there is no need for a large U to soften the phonons or translating the spectra by hundreds of meV to match of DFT and ARPES. Indeed, one may think that the broadening of the bands is a consequence of the electronic disorder driven by the dimerization of trigonal Ge, hence making the problem hard at the very quantitative level. Nonetheless, this does not discard the

presence of electronic correlations; there must be since the compound is magnetic, but diminish their importance in regards to the periodic lattice distortions developed at low temperature. Unfortunately, we do not have a large energy window to obtain the bandwidth and compare with DFT. Samples were very difficult to cleave and we only focus on obtaining the best energy/momentum resolution close to the Fermi level.

Some even more minor points to consider: It is stated that “surface state pocket emerging at Gamma’ of the neighboring BZs”. Can the authors label on the map which is the surface state? While the DC1, 2 and 3 are all labeled in the measured data, could the corresponding features be also labeled in the calculated dispersion in Fig. 2B as well? Naively when I compared these two side by side, while the number of Dirac cones and even possibly the dispersion slopes are reproduced by calculations, the locations do not appear to match. It would help the readers if the corresponding features are labeled for a better comparison.

- We have marked the surface state that appears in the DFT surface calculation and updated the figure 2 accordingly.

In similar veins, for the ease of the readers who would like to follow the discussion of the measured dispersions on pages 4 and 5, it may be helpful to refer to some of the bands with labels such as alpha, beta, etc, directly labeled on the figures.

- In Figure 2(E), we have labeled the bands with Greek letters for a better discussion in the rest of the paper.

Figure caption for Fig. 1: H and I should be combined according to the figure. Figure caption for Fig. 3B: the blue ellipse mentioned seems to be missing in the figure. Fig. S19: the caption says both annealed and as-grown samples are shown. But I’m not sure if the figure shows both, or only the annealed crystals since the temperature are all different? For the FeGe_{0.9} sample shown in the SM, what is the transition around 200K from the magnetization?

- We thank the referee for spotting these typos. We have combined the captions for both Fig. 1H and 1I and added a blue ellipse that was missing in panel 3B. In fig. S19, the top plot shows the annealed and the bottom the as-grown FeGe crystals.

We haven't paid too much attention to the FeGe_{0.9}, as extrinsic disorder complicates the interpretation of the DS based on order/disorder models. Nevertheless, we suspect that the transition around 200K might be related with the B20 phase of ferromagnetic FeGe, which has cubic symmetry and holds a skyrmion phase at high temperature. This is in line with the sample annealed at 550 K that shows a transition characteristic of cubic FeGe, see for instance Phys. Rev. Lett. 108, 267201 (2012).

Reviewer #2 (Remarks to the Author):

In the manuscript by Subires and colleagues, they studied the charge density wave (CDW) ordered phase in the antiferromagnetic kagome metal FeGe using x-ray diffraction (XRD), angle-resolved photoemissions spectroscopy (ARPES), density functional theory (DFT), and diffuse scattering. They first characterized lattice parameters as a function of temperature by XRD, identifying two characteristic temperature scales, namely T* and TCDW. After the identification, they defined a relevant order parameter, the fraction of dimers given by the occupancy of corresponding Ge1 sites. Next, they carried out ARPES experiments and compared experimentally obtained electronic structures with their DFT calculation results. These results showed a good match, and the authors claimed that correlation effects may not be significant, contrary to previous studies. Finally, they presented diffuse scattering results at various temperatures together with anisotropy in diffuse scattering, showcasing evidence of short-range positional but quasi-long-range band orientation order. Ultimately, they described the nature of the CDW order in FeGe as an order-disorder scenario which follows a second-order like phase transition.

This is a comprehensive study, combining diffraction, diffuse scattering, and spectroscopy with theoretical calculations. The Supplementary Information is also extensive, providing valuable experimental and theoretical information on FeGe.

- We acknowledge the referee for the critical reading of the manuscript and his/her positive feedback.

However, they still need to confirm their main arguments with recent experimental reports, particularly on annealed FeGe. While they presented some of data sets from 300 °C-annealed FeGe in the Supplementary Information, more data are needed. I would like to ask authors to include datasets from various annealed FeGe, for examples, annealing temperature variation as shown in Wu et al., Phys. Rev. Lett. 132, 256501 (2024). Before I decide whether to recommend this manuscript for publication in Nature Communications, I would like to confirm if authors' scenario remains valid after these additional verifications. I outlined some critical points (leaving minor points aside for now) below.

- We thank the referee for his/her comment. Following the question of the referee 1 (see response above), we have annealed samples at 440°C and 550°C. The new data and its discussion are now part of the supplementary information.

1. They mentioned that normalized CDW peak intensity from XRD showed a linear T-dependence down to 80 K, implying that the CDW transition cannot be first-order, further supported by the small release of specific heat. However, a recent study on annealed FeGe, which has the long-range CDW order, shows significant peak structure at T_{CDW} in the specific heat characterization (arXiv:2307.07990), strongly supporting a first-order like nature of the CDW transition. How can this discrepancy be resolved?

- Our as-grown FeGe crystals show a very small anomaly in the specific heat. We believe that this is derived from the order-disorder transformation of dimerized and undimerized phases, similar to the order-disorder phase transitions observed in spin glasses, see for instance (J. Phys.: Condens. Matter 7 4193 (1995) and more recently Phys. Rev. Materials 4, 064412 (2020), where the phase transition is also continuous. Once the sample is annealed at 300 °C, the frustration induced by the dimerized/undimerized disorder (not for all the annealing temperatures as we show now in the SI) is released and the phase is more 'first order-like', see figure below. This agrees with the data of Chen et al (arXiv:2307.07990). Note that further annealing (400°C) makes the transition a bit more continuous again (see response to the first referee and the DS data of the new annealed samples in the SI), complicating the analysis of the phase transition.

We agree with the referee that the definition of first or second order phase transition, which require the analysis of the critical exponents, would lead to discrepancies and misunderstandings, hence we have preferred to rename it as continuous or discontinuous phase transitions.

Temperature dependence of the CDW in FeGe after annealing at different temperatures (see inset).

2. In their ARPES results, they did not find any signatures of a gap opening, which contradicts with the preprint on annealed FeGe (arXiv:2404.0223). This points also needs to be resolved. Furthermore, termination dependence (i.e., kagome & honeycomb) should be examined to fully address surface-originated bands in ARPES.

- The referee is correct, and we would like to thank him/her for this comment (also pointed out by other referee), as it let us to solve this inconsistency and rephrase this part of the manuscript. All the samples we measured in ARPES (dozens of crystals) cleaved in the honeycomb terminated surface. Unfortunately, we do not have data with the kagome termination. In any case, what we aim to say in the manuscript is that we did not observe any gap within the 20 meV energy resolution limit of our ARPES measurements. Consequently, as we are aware of authors reporting gaps by either STM and ARPES, we decided to delete such statement and have rephrased the text accordingly.

3. The authors downplayed the role of correlation effects based on good matches between the ARPES and DFT results with accurate consideration on the space group. I find this statement unconvincing, as they also mentioned, “Besides the aforementioned features that have a good matching between DFT bands and ARPES, we also observe some features that cannot be well matched.” Those two statements are confusing due to their self-inconsistency. In addition, considering renormalization factors for DFT calculations are quite common for transition-metal-based systems (e.g., Fe-based superconductors), supported by comparisons between experimental angle-integrated valence band spectra and density of states calculations. It is hard to believe that FeGe is uniquely exempt from this general behavior of transition-metal-based systems.

- We thank the referee for his/her comment, which goes in line with the first referee. What we mean is that we can achieve a reasonable match between ARPES and DFT without the need of renormalization factors or U. Indeed, the DFT calculations carried out before did not consider either the surface terminations or the surface states, which, we believe, are the most important to first understand the fermiology of kagomes. We think that the lack of a *perfect* match between our ARPES and DFT is a consequence of the inaccurate treatment of dimerized and undimerized phases in DFT and the level of electronic disorder that this implies and multiorbital character of FeGe and kagome metals in general.

We have reformulated the sentence ‘*Besides the aforementioned features that have a good matching between DFT bands and ARPES, we also observe some features that cannot be well matched*’ by ‘*The slight disagreement between our ARPES data and the DFT calculations could have its origin in the improper treatment of dimerized and undimerized phases in DFT and the multiorbital character of the Fe*’

Reviewer #3 (Remarks to the Author):

I have revised the present paper submitted to Nature Communications. The work contains a thorough analysis of the CDW transition in FeGe regarded as a Kagome metal. The authors elucidate the origin of such transition which they ascribe to interlayer Ge-Ge dimerization occurring below TCDW. They characterize the transition by introducing an Ising model describing the Kagome lattice degrees of freedom (dimerized vs. non-dimerized sites). One strength of the paper which I highly value is the combination of several experimental (X-ray diffraction, diffuse scattering, ARPES) and theoretical techniques (DFT, Monte Carlo techniques) on the same material. In this way the authors are able to reach a solid conclusion on the origin of the CDW transition observed. Since there is no consensus yet on the mechanism driving the CDW phase in FeGe, I believe that the present paper is timely, interesting and the observations reported relevant to understanding the physics of Kagome metals in general.

- We again acknowledge the referee for his/her enthusiastic reception of our work and his/her positive comments. In the following, we are giving a response to his/her comments/questions.

Nevertheless, I believe that some issues can be tightened further in the paper before being published. For instance, it would be useful if the present scenario for the CDW transition is more readily set in context comparing with previous descriptions. In particular, I miss the link with topological phenomena reported previously. More detailed comments can be found below.

Main issues/comments:

-There are indications from previous experimental work (Teng et. al., Nature 609, 490 (2022)) of an anomalous Hall effect occurring within the CDW state when $T < T_{CDW}$. I am wondering how relevant is this to the CDW formation from your experimental observations. Do you find any indications of such AHE? If that is the case how do you think model (1) should be extended to take account of such phenomena?

- We have also measured similar AHE (a deviation from the linear behavior) as reported by Teng et al. in our as-grown FeGe crystals, see figure below, although, we are not commenting whether this AHE is a signature of the chiral flux (or loop current in kagome layer).

Magnetic field dependence of the transverse resistivity

However, we believe that the AHE observed here is related with non-trivial topology of the Fermi surface and not with the topological character of the phase transition, as stressed by the referee in his/her last question, which we answer in detail below. Furthermore, the Ising model is developed to describe the order-disorder transformation, namely, the ordering with propagation vectors $(0\ 0\ 1/2)$, $(1/2\ 0\ 0)$ and $(1/2\ 0\ 0\ 1/2)$ but not the topological character of the phase transition. This topological description comes from the KTHNY model and the melting of CDW by dislocations and/or disclinations, which, according to our knowledge, has no connection to the Berry curvature of FeGe.

-One important message from the paper seems to be the small role played by electronic correlations in the CDW transition observed in FeGe. However, the large room temperature values of resistivity observed in FeGe as compared to those of AV3Sb5 (Teng et. al., Nature 609, 490 (2022)) seem to suggest that FeGe can be regarded a bad metal (or close to a bad metal). Such bad metal behavior is characteristic of strongly correlated materials. Isn't there then an inconsistency between your conclusions and the resistivity observations?

- We thank the referee for his/her question. Indeed, he/she is correct in his/her arguments. However, in FeGe, we believe that the large value of the resistivity at RT is just a consequence of the fluctuating disorder of dimerized/undimerized phases that affects the electron scattering. Note that our DS measurements demonstrate a disordered matrix composed of dimerized Ge atoms in the kagome plane and in-plane Fe displacement above 300 K.

-I miss an explanation of how model (1) is derived. For instance, which is the relation between the c_i and microscopic quantities of the material?

- The model (1) is the Hamiltonian used in the Monte Carlo simulation for modeling the diffuse scattering. It is based on the Ising model of adjacent spins and is commonly used to describe binary substitution disorder with the 'spin' variables corresponding to one of two possible atoms occupying a lattice

site. The simplest Ising model considers only the nearest neighbour interactions and an overall dipole. Empirically, we found that 3 nearest neighbour shells in the ab plane and just the nearest neighbour shell along the c direction were required to reproduce the diffuse scattering.

The strength and direction of interacting spins is controlled through the c parameters. They represent an interaction strength or force constant between two spins along a particular direction; their size and sign therefore indicate the relative strength of interactions in different neighbour shells to drive the MC simulation towards a configuration that reproduced the diffuse scattering. In this work, the values were based on DFT energy calculations. We have updated the SI part.

-The point is made that the CDW is caused by the dimerization between Ge atoms on different layers. It is not clear to me if this transition is actually driven by the electrons which order in the Kagome lattice planes which then can lead to the Ge dimerization observed. In such case the Coulomb repulsion between electrons on the Kagome lattices may drive the CDW instability. And then as a consequence of this the lattice distortion may occur. Could this scenario be also consistent with your observations?

- We appreciate the comment of the referee. Indeed, his/her argument is correct and goes to back to understanding the origin of the CDW transitions, which most of the times is a 'chicken and egg' problem. According to our results and the literature in FeGe (Nature Communications 14, 6183 (2023).), we believe that the CDW instability is a consequence of a 'magnetostriction' effect that takes place at the Neel temperature (~400 K) that induces a shift of the trigonal Ge atoms in the kagome plane (the so-called dimerization). This has been called 'magnetic energy saving' in the literature (Phys. Rev. Mater. 7, 104006 (2023).). As we cool down to T_{CDW} , the gain of magnetic energy increases until the CDW takes place at 105 K. The first consequence is the renormalization of the p_z orbital of trigonal Ge at Γ . Therefore, we think that the electronic ordering happens as a consequence of the dimerization and not the other way around. In any case, the referee raised a good point, which perhaps could only be solved if one is able to track the temperature dependence on the electronic band structure from 400 K down to T_{CDW} with sub-meV energy resolution.

-STM experiments (J.-X. Yin, et. al., PRL 129, 166401(2022)) display a suppression of the differential conductance at the Fermi energy when entering the CDW for $T < T_{CDW}$. However, your ARPES data indicate that no gap opening occurs at the CDW transition. Can you explain the two seemingly different observations?

- We thank the referee for this question, as outlined above by the referee (1). We do not have the energy resolution as STM, thus our data cannot discard the opening of a gap. Our best energy resolution was ~20 meV and FeGe bands are intrinsically broad. We have rephrased this sentence to avoid confusion and/or misunderstanding.

-There is also evidence for the existence of an edge state in the CDW phase as observed in STM measurements (J.-X. Yin, et. al., PRL 129, 166401 (2022)). This edge state is of topological origin and is related to the topological character of the CDW state (and the AHE observed). The edge state is found to disappear at temperatures $T > T_{CDW}$. Is your analysis consistent with the existence of such edge state? Could it be related to the absence of a gap in the CDW as observed in ARPES?

- Although our analysis of the phase transition seems to agree with temperature dependence of the observed edge states, we think that these are different things. The presence of edge states and relation to the AHE is a consequence of the non-trivial topology of the band structure, namely Berry curvature. In our case, we refer to a topological transition that is driven by the melting of pairs of dislocations/disclinations, as described by the KTNHY theory. We do not observe a gap in ARPES within our energy resolution, but such gap is reported by STM. Indeed, the referee's comment about the presence of edge states would go more in line with the STM arguments than with the diffraction data we present. From our results, we cannot make a clear connection with the AHE. Besides, to avoid wrong claims, we have rephrased the parts of the text about any signature of gap opening.

We thank the referee's comments and his/her feedback that allowed us to add more information to the SI, discuss and update the bibliography and improve the communication of our results.

Reviewer #2 (Remarks to the Author):

In the revised manuscript by Subires and colleagues, they have addressed almost all the concerns raised in the previous report. However, the response regarding annealed FeGe remains somewhat lacking, as they have not provided specific heat dataset in the response. Given that inconsistency in specific heat results could mean significant sample dependence, an explicit dataset for specific heat must be presented.

We thank the referee for his/her comment. We have measured the specific heat and the magnetization of the 300°C annealed sample. The data is presented in the SI Fig. S18. As expected, the C_p shows a sharp step anomaly from a transition at $T_{CO} = 110$ K, clearly of weak 1st order character and consistent with the reports in the literature. Following the recent works with annealed FeGe, we are carrying out IXS experiments in the next weeks at ESRF in the annealed samples to study the spin-phonon mechanism as in H. Miao et al., Nature Comm. 14, 6183 (2023).

Additionally, during the review process, a preprint (arXiv:2410.13994) has been available online, providing experimental evidence on how annealing affects FeGe crystals. From their 4D-STEM results, Ge deficiency could be responsible for the absence of the CDW order in annealed samples at 550~560 degs. Among the authors' results from FeGe_{0.9}, the magnetic susceptibility closely resembles that of FeGe annealed at 550~560 deg. However, as they assigned a CDW transition for FeGe_{0.9}, and I wonder if they could provide a consistent explanation for the new results. The last note is that more papers have been available online during the review (arXiv:2410.18063, arXiv:2411.10931, and arXiv:2412.18824), and some of them might be relevant and worth to be included in the manuscript.

We thank the referee for spotting this and providing the newest literature. Indeed, we haven't noticed the similarity in the magnetization between FeGe_{0.9} and FeGe annealed at 550°C. In this sense, we have also reshaped the explanation of the magnetization of FeGe annealed at 550°C following the arguments of Klemm et al., since after a careful analysis of the x-ray data, we haven't detected the cubic (skyrmion) phase of FeGe. On the other hand, our explanation for FeGe_{0.9} was merely a suggestion, namely disorder, which is a very broad scenario. Indeed, the data presented in the arXiv:2410.13994 seem to provide a more detailed understanding of the physics of Ge-deficient FeGe. Nevertheless, we do see a short-range charge order FeGe_{0.9} at 100K, which we will try to clarify in the next beamtimes. We have reshaped the discussion of FeGe_{0.9} accordingly, cited the manuscript in the main text and updated the bibliography.